# Colorectal Cancer Chemoprevention: A Dream Coming True?

**DOI:** 10.3390/ijms24087597

**Published:** 2023-04-20

**Authors:** Martina Lepore Signorile, Valentina Grossi, Candida Fasano, Cristiano Simone

**Affiliations:** 1Medical Genetics, National Institute of Gastroenterology—IRCCS “Saverio de Bellis” Research Hospital, Castellana Grotte, 70013 Bari, Italy; martina.lepore@irccsdebellis.it (M.L.S.); candida.fasano@irccsdebellis.it (C.F.); 2Medical Genetics, Department of Precision and Regenerative Medicine and Jonic Area (DiMePRe-J), University of Bari Aldo Moro, 70124 Bari, Italy

**Keywords:** colorectal cancer, chemoprevention, precancerous lesions, target populations

## Abstract

Colorectal cancer (CRC) is one of the deadliest forms of cancer worldwide. CRC development occurs mainly through the adenoma-carcinoma sequence, which can last decades, giving the opportunity for primary prevention and early detection. CRC prevention involves different approaches, ranging from fecal occult blood testing and colonoscopy screening to chemoprevention. In this review, we discuss the main findings gathered in the field of CRC chemoprevention, focusing on different target populations and on various precancerous lesions that can be used as efficacy evaluation endpoints for chemoprevention. The ideal chemopreventive agent should be well tolerated and easy to administer, with low side effects. Moreover, it should be readily available at a low cost. These properties are crucial because these compounds are meant to be used for a long time in populations with different CRC risk profiles. Several agents have been investigated so far, some of which are currently used in clinical practice. However, further investigation is needed to devise a comprehensive and effective chemoprevention strategy for CRC.

## 1. Introduction

Colorectal cancer (CRC) is the second deadliest cancer and the third most common malignancy worldwide [1]. Based on recent global trends, the incidence rate of CRC is consistently growing and is expected to increase by 60% until 2030, reaching more than 2.2 million new cases and 1.1 million deaths [2]. CRC carcinogenesis is a multiyear, multistep, and multipath process characterized by progressive genetic alterations and associated tissue damage [3]. More than thirty years ago, Fearon and Vogelstein proposed the adenoma-carcinoma sequence model for CRC development, suggesting that decades are needed for progression from adenoma to carcinoma and eventually to metastatization (Figure 1A). Consistent with this model, histopathological and molecular data showed that almost all colorectal carcinomas arise from adenomas, which continuously progress through increases in size, dysplasia, and the acquisition of villous morphology [4]. Currently, it is widely accepted that there are four different stages of colorectal tumorigenesis: early adenoma, late adenoma, carcinoma, and metastatic cancer [4]. 

At the genomic level, colorectal tumorigenesis occurs through a sequence of genetic alterations involving tumor suppressor genes (loss of APC at 5q, DCC at 18q, and TP53 at 17p) and oncogenes (such as mutations in KRAS at 12p) [4], which provides a framework for the study of this complex process. Indeed, splitting up CRC development into separate steps offers a window of opportunity for effective prevention [5]. While fecal occult blood tests and colonoscopy remain the gold standard for CRC prevention [6], interest in chemoprevention has been growing in recent years. The term chemoprevention was first coined in 1976 [7] and was defined as the use of a natural or synthetic substance to delay the time of cancer onset, reverse the process of carcinogenesis or prevent tumor recurrence and metastasis (Figure 1B) [8,9]. Nevertheless, finding compounds capable of successfully preventing CRC turned out to be very difficult [9], and new agents are needed to treat clinically evaluable precancerous lesions as well as the entire colon epithelial sheet at risk [10].

## 2. Precancerous Colorectal Lesions

The identification of significant endpoints that can be used in clinical trials is crucial to devise an effective chemoprevention strategy. During the early steps of CRC tumorigenesis, colon epithelial cells begin to show uncontrolled proliferation, which results in the formation of aberrant crypt foci (ACF), followed by the development of a polyp and, subsequently, an adenoma (Figure 1A). ACF are thus the earliest neoplastic lesions in CRC carcinogenesis and can be hyperplastic or dysplastic [11]. Hyperplastic ACF are larger and longer than adjacent normal colonic mucosa, and their luminal opening is serrated and slightly elevated from the surrounding tissue. On the other hand, dysplastic ACF are characterized by hypercellularity with abnormal nuclear features such as hyperchromatism and loss of polarity. Moreover, dysplastic ACF stains are positive for PCNA and Ki67 more extensively at the upper and middle crypt compartments compared to hyperplastic ACF and are believed to be precursor lesions of adenomas [11].

As a further step in the adenoma-carcinoma sequence, cells with high WNT activity emerge from ACF and evolve into a tubular or tubule-villous polyp. The uncontrolled proliferation of a polyp subsequently leads to the development of an early adenoma [12]. These commonly asymptomatic lesions are frequently found during colonoscopy screening [13]. The transformation rate of adenomas to carcinomas is about 0.25% per year. The size of the adenoma is crucial in this process as cancer arises from 1% of adenomas < 1 cm, 10% of adenomas between 1 cm and 2 cm, and 50% of adenomas larger than 2 cm. Moreover, the malignant potential of an adenoma is also related to its growth pattern and grade of dysplasia [13]. 

It is widely recognized that the risk of developing invasive cancer increases with the progression of precancerous lesions, even if not all ACF or adenomas progress to CRC [10]. Interestingly, there is a clear correlation between molecular mutations and histological phenotype. Indeed, mutation of APC was determined to be an early event occurring before the development of an adenoma, while mutation of TP53 was identified as a late event denoting the transition from adenoma to carcinoma [10].

## 3. Target Populations for Chemoprevention

Identifying target populations at average, moderate, and high risk of developing adenomas and hence CRC is an important step in the investigation and implementation of chemopreventive agents (Figure 1B), allowing candidate compounds to be tailored to the specific risk level of affected individuals.

In the general population, which has an average risk of developing CRC, important factors that affect risk calculation are sex, age, and race/ethnicity [14]. CRC rates are 30% higher in men than in women, and the risk increases with age. Moreover, among broadly defined racial and ethnic groups, CRC incidence is highest in non-Hispanic Black individuals. Reasons for racial/ethnic disparities in CRC are complex but largely related to socioeconomic status. In addition, approximately half of all CRCs are attributable to lifestyle factors, including unhealthy diet, high alcohol consumption, smoking, and lack of physical activity [15]. 

Populations with a moderate risk of developing CRC include subjects with a prior diagnosis of colonic adenoma and individuals with a family history of CRC [9]. Indeed, people with a first-degree relative diagnosed with CRC have up to four times higher risk of developing the disease compared with people without a CRC family history [16].

Populations at high risk of CRC consist of patients with CRC-related genetic syndromes. Almost 5% of CRC patients have a germline mutation associated with high-risk genetic syndromes [17]. The most common hereditary risk factor for CRC is Lynch syndrome, also known as hereditary non-polyposis CRC (HNPCC). HNPCC is due to inherited mutations in genes affecting DNA mismatch repairs, such as MLH1, MSH2, MSH6, PMS2, and EPCAM [18]. The second most common genetic syndrome predisposing to CRC is familial adenomatous polyposis (FAP) [13]. FAP is characterized by the development of up to thousands of polyps until the third decade of life [19] and is related to germline mutations in the APC gene. There are various types of FAP: classic FAP, attenuated FAP (AFAP), which is a less severe form of the disease, and other rare variants such as Gardner’s syndrome, which is accompanied by extra-intestinal manifestations, and Turcot’s syndrome, which is associated with brain tumors [20]. Recently, another type of FAP has been described, which was termed gastric polyposis and desmoid FAP (GD-FAP). This syndrome is characterized by colon oligo-polyposis, diffuse gastric polyposis, and desmoid tumors and is related to a truncation mutation in the C-terminal region of the APC gene [21]. Moreover, other genetic syndromes predispose to CRC, including MUTYH-associated polyposis, Peutz-Jeghers syndrome, juvenile polyposis syndrome, Cowden syndrome, and hamartoma tumor syndrome, all of which are rare diseases with an up to 40% increased lifetime risk of CRC [22,23,24,25,26,27,28]. 

Inflammation is also linked to CRC carcinogenesis. Indeed, patients with chronic inflammatory bowel disease such as ulcerative colitis or Crohn’s disease are at greater risk of developing CRC. Available reports suggest that almost 15% of patients with a 30-year history of ulcerative colitis will develop CRC [29].

## 4. Chemopreventive Agents

Since chemopreventive agents are meant to be administered for a long time, especially in the average-risk population, they should be well tolerated and have low side effects. Moreover, they have to be readily available at a low cost and easy to administer, with a convenient dosing schedule for patients. Candidate compounds must thus fulfill these requirements before their efficacy can be evaluated in a clinical trial [9]. In the last decades, several natural and synthetic molecules have been investigated as potential chemopreventive agents for CRC. These agents and the relevant studies are described in detail below and summarized in Table 1. 

### 4.1. Anti-Inflammatory Agents

Since inflammation can promote the onset of CRC, it is reasonable to assume that agents with anti-inflammatory activity may have chemopreventive effects.

#### 4.1.1. Aspirin

The most widely studied non-steroidal anti-inflammatory drug (NSAID) for CRC chemoprevention is acetylsalicylic acid. This compound was marketed as aspirin in 1899 [30] and is currently the most promising chemopreventive agent for CRC. The mechanistic basis for its protective effect is believed to be its irreversible binding to, acetylation, and consequent inhibition of PTGS1 and PTGS2, also known as cyclooxygenase 1 (COX-1) and 2 (COX-2), which ultimately results in prostaglandin E2 (PGE2) downregulation [31]. Indeed, COX enzymes are responsible for the conversion of arachidonic acid into downstream effectors that are metabolized into prostaglandin and eicosanoids. Consistently, increased expression of COX-2 has been found in up to 40% of colonic adenomas and up to 90% of sporadic CRCs [32,33], and increased synthesis of PGE2 has been observed in patients with CRC and has been shown to promote CRC carcinogenesis [34]. 

##### Evaluation of Aspirin Treatment with ACF or Adenoma Lesions as Endpoints

In vivo experiments showed that aspirin treatment was associated with reduced ACF lesions, which are the smallest lesions detectable in normal-appearing human colonic mucosa. The administration of 0.2% or 0.6% aspirin in rats treated with the carcinogenic drug azoxymethane promoted a 55% and 54% reduction, respectively, in the overall frequency of ACF (crypts/focus). In particular, aspirin seemed to lower the frequency of medium and large ACF but not of the small ones. These findings show that aspirin acts by delaying the initiation of azoxymethane-induced CRC carcinogenesis in rats and suggest that it has a chemopreventive effect on ACF [35,36]. 

Several randomized controlled trials (RCTs) revealed the efficacy of aspirin chemopreventive treatment on colorectal adenomas [37,38,39,40]. From 1993 to 2000, the Cancer and Leukemia Group B (CALGB) enrolled individuals with prior CRC and treated them with aspirin for several years. Investigators observed that these patients had a lower risk of subsequent colonic adenomas and developed adenomas at later stages [38]. Aspirin chemoprevention was also studied in FAP patients as a high-risk group. The Colorectal Adenoma/Carcinoma Prevention Program (CAPP1) study, conducted from 1993 to 2005, found that aspirin did not reduce colonic polyps in these patients, but one year or more of aspirin use decreased the largest polyp size [41]. The Aspirin/Folate Polyp Prevention Study (AFPPS) enrolled 1121 patients with a recent history of adenomas as a moderate-risk population from 1994 to 1998. In this group, low-dose aspirin had moderate chemopreventive activity on adenomas in the large bowel [37]. Another study, the United Kingdom Colorectal Adenoma Prevention (ukCAP) trial, was conducted from 1997 to 2005 to evaluate the chemoprevention of polyp formation in the bowels. This large study involved 945 patients with moderate CRC risk who already had one or more polyps removed. The results showed that aspirin (300 mg/day) but not folate (0.5 mg/day) treatment reduced the risk of colorectal adenoma recurrence and prevented the development of advanced lesions [39]. 

In a randomized, double-blind, placebo-controlled trial carried out from 1997 to 2001 by the Association Pour la Prevention Par l’Aspirine du Cancer Colorectal (APACC), two different endpoints, i.e., adenoma recurrence after one and four years, were analyzed. Daily soluble aspirin was associated with a reduction in the risk of recurrent adenomas, as detected by colonoscopy one year after starting treatment [40]. However, these promising results were not confirmed after four years. Two possible explanations have been proposed for this discrepancy. The first reason could be the lack of statistical power of the final analysis performed at four years, resulting from the number of dropouts (about 30%) mainly due to the long duration of the follow-up and the need for a third colonoscopy. The second reason could be a differential effect of aspirin according to the exposure time and the natural history of polyps. Indeed, the authors distinguished between the “true” chemopreventive aspirin activity, which is only observed after 7–10 years of treatment, and its anti-tumor effect detected at 1 year [42]. 

The Systematic Evaluation of Aspirin and Fish Oil (seAFOod) Polyp Prevention Trial was a multicenter, randomized, double-blind, placebo-controlled study conducted from 2010 to 2017 on moderate-risk patients aged 55–73. The first endpoint was the detection of adenomas during the first-year surveillance colonoscopy. Unfortunately, neither of the two study treatments, i.e., eicosapentaenoic acid (EPA) and/or aspirin, were associated with reduced adenoma rates [43]. Importantly, clinical trials commonly assess more than one outcome. Usually, secondary endpoint results improve the overall interpretation of the trial and facilitate the understanding of the extent of any possible intervention effect [44]. A secondary analysis of the seAFOod trial provided evidence of the chemopreventive effects of both agents. In particular, aspirin was effective in reducing the number of conventional or serrated adenomas in the right colon [43]. Of note, a large cross-sectional study performed from 2011 to 2014 revealed that adenoma prevention by aspirin treatment is abrogated in active smoker patients [45]. 

Overall, the studies described above provided promising results, showing that aspirin treatment can reduce adenoma risk, size, and recurrence.

##### Evaluation of Aspirin Treatment with CRC as an Endpoint

A large body of evidence suggests that regular prophylactic aspirin use reduces CRC incidence and mortality both in the general average-risk population and in high-risk groups consisting of patients with CRC-related genetic syndromes. 

The first prospective cohort study with CRC as an endpoint was carried out from 1980 to 2000. This study, named the Nurses’ Health Study (NHS), enrolled 82,911 women and demonstrated that regular long-term aspirin use reduces the risk of CRC [46]. However, these findings were not always confirmed in subsequent studies.

The Physicians’ Health Study (PHS), which was conducted from 1982 to 1995, found that alternate-day use of 325 mg of aspirin had no statistically significant effect on the incidence of CRC after up to 12 years of follow-up [47]. Conversely, the Cancer Prevention Study II (CPS-II), a prospective mortality study that was also started in 1982 and included 662,424 adults, showed that regular aspirin use at low doses may reduce the risk of fatal CRC [48]. The prospective cohort study Health Professionals Follow-up Study (HPFS) began in 1986 to determine whether the regular use of aspirin decreases the risk of CRC. Approximately 47,900 male subjects with an age ranging from 40 to 75 years completed the questionnaire. In their report about the study, the authors highlighted that regular users of aspirin (more than two times per week) had a lower risk of CRC and metastatic CRC [49]. However, this association was not confirmed 4 years after the suspension of aspirin administration [50].

Subsequently, the Women’s Health Study (WHS), a large placebo-controlled randomized trial that was started in 1993 and lasted 20 years, first failed to confirm that aspirin protects against CRC. It was initially speculated that these findings could be related to the low doses of aspirin used (100 mg every other day) or to the insufficient duration of the treatment; however, it was more probably due to the short duration of the follow-up [51,52]. Indeed, an inverse association between the use of aspirin and CRC incidence emerged 10 years after patient randomization [52], and the 18-year follow-up of WHS patients confirmed a reduction in CRC risk in patients administered with alternate-day low-dose aspirin [52]. 

Interestingly, the 1997–2012 analysis based on the Colon Cancer Family Registry (CCFR) showed that the use of aspirin decreased CRC risk in Lynch syndrome patients [53]. CRC incidence in this patient population was further investigated in the Colorectal Adenoma/Carcinoma Prevention Program 2 (CAPP2), which enrolled a high-risk cohort of patients affected by Lynch syndrome from 2001 to 2008. While the analysis of the primary outcome performed after two years of follow-up failed to show a reduction in CRC risk [54], the analysis of the secondary outcome carried out after 10 years of follow-up revealed a remarkable decrease in the risk of developing CRC [55]. 

In 2004, the US Preventive Services Task Force (USPSTF) recommended for the first time the use of aspirin as a chemopreventive agent for the prevention of both cardiovascular disease and CRC in non-high-risk populations. However, in 2015, it specified that the daily use of aspirin was indicated in patients between 50 and 69 years of age with a specific cardiovascular risk profile (10-year risk of cardiovascular disease, expected to live more than 10 years without increased bleeding risk) [56,57].

The Japan Colorectal Aspirin Polyps Prevention (J-CAPP) study, conducted from 2007 to 2012, revealed that aspirin reduced the recurrence of adenoma or CRC in the non-smoker population [58]. 

In 2008, the ASCOLT RCT started enrolling patients with high-risk Dukes’ B and C CRC. Its primary endpoint was disease-free survival, while its secondary endpoint was five-year overall survival. Patients who had undergone surgery and chemotherapy treatments (oxaliplatin) were assigned to daily use of 200 mg aspirin or placebo for 3 years [59]. Follow-up assessments were performed every 3 months for 3 years and then every 6 months for another 2 years. To date, no results have been provided for this study. 

In 2010, Rothwell and colleagues published the results of a follow-up analysis of pooled individual patient data from four randomized trials of aspirin versus control and one trial of different doses of aspirin [60]. The authors found that aspirin taken for several years at doses of at least 75 mg daily reduced long-term CRC incidence and mortality. Moreover, the greatest benefit was observed for cancers in the proximal colon, which are not effectively prevented by colonoscopy [60].

The Aspirin Intervention for the Reduction of CRC Risk (ASPIRED) trial was launched in 2010 to define which patients can benefit more from aspirin use [61]. This double-blind, multidose, placebo-controlled study was addressed to patients previously diagnosed with colorectal adenoma. 180 patients (60 per arm) were randomized to low-dose (81 mg/day) or standard-dose (325 mg/day) aspirin or to placebo [61]. This trial is still recruiting, and no results have been published yet. 

As mentioned above, the CAPP2 study tested the effect of the daily use of high-dose aspirin (600 mg per day) in patients with Lynch syndrome. Some years later (from 2014 to 2019), the CAPP3 trial was conducted on more than 1500 Lynch syndrome patients to establish the most effective aspirin dose in this population. Patients were treated with 100 mg, 300 mg, or 600 mg of aspirin. The first results of this study are expected by the end of 2024 [62].

A randomized, double-blind, placebo-controlled, multicenter trial, named J-FAPP, was carried out in Japan between September 2015 and March 2017 on another population at high risk of CRC. This study enrolled patients with FAP to evaluate possible alternatives to colectomy as a preventive treatment [58]. Patients were divided into four treatment groups: daily aspirin (100 mg) plus daily mesalazine (2 g), daily aspirin (100 mg) plus mesalazine placebo, aspirin placebo plus daily mesalazine (2 g), or aspirin placebo plus mesalazine placebo. Treatment was continued until 1 week before the 8-month colonoscopy. The results showed that daily use of low-dose aspirin reduced the recurrence of colorectal adenomas larger than 5 mm and CRC in FAP patients [58].

Current guidelines support the recommendation of 100 mg/day aspirin to reduce CRC risk only for Lynch syndrome patients [63,64,65]. In the next few years, the findings of various ongoing trials on aspirin in CRC or in other settings, e.g., Aspirin to Reduce Risk of Initial Vascular Events (ARRIVE) and Aspirin in Reducing Events in the Elderly (ASPREE), are expected to add evidence in support of aspirin use in CRC chemoprevention.

The available data suggest that for aspirin to have chemopreventive effects, it should be taken for 10–20 years and at doses greater than those used for cardiovascular prevention. However, the optimal dose for specific patient groups, the exact duration of the treatment, and when it should be started still need to be elucidated. Moreover, benefits should be balanced against potential harms. Indeed, side effects of aspirin treatment include gastrointestinal tract bleeding and intra/extracranial hemorrhages, mostly in patients older than 70 years [66]. In this respect, it should be noted that eradication of Helicobacter pylori infection before regular use of aspirin may reduce the incidence of gastrointestinal complications by 25–30% [67]. 

##### Biomarkers for Aspirin Efficacy as a CRC Chemopreventive Agent

Over the last few years, several biomarkers have been investigated to help identify which CRC patient groups can benefit the most from aspirin chemoprevention. Indeed, altered levels of selected molecules, specific mutations, and certain genetic variants have been shown to affect the efficacy of aspirin use as a CRC chemopreventive treatment. These biomarkers are summarized in Table 2.

Interestingly, varying genetic susceptibility among patients enrolled in clinical trials and among different CRC subtypes has emerged as an important factor that may influence trial results. Indeed, an explanation for the conflicting results of some of the trials on aspirin chemoprevention in CRC may be the presence of genetic variants in genes affecting aspirin activity.

Uridine diphosphate glucuronosyltransferase 1A6 (UGT1A6) is involved in aspirin metabolism by catalyzing its glucuronidation, and two UGT1A6 single-nucleotide polymorphisms (SNPs), rs2070959-G and rs4365457-C, have been shown to be associated with 30–50% lower enzyme activity compared with the wild type [68]. Of note, in studies comparing the effect of regular aspirin intake on adenoma risk in subjects with wild-type or variant UGT1A6 genotypes, the benefit was largely confined to the groups carrying these functional polymorphisms, while in subjects with wild-type UGT1A6, aspirin use was not associated with a statistically significant reduction in the risk of adenoma [69,70].

Recently, a subgroup of patients enrolled in the AFPPS trial was investigated with a genotyping approach. As a result, two novel SNPs, rs2430420-GG and rs28362380-TT, which seemed to be potential markers of daily low-dose aspirin (81 mg) efficacy, were identified in the promoter of the ornithine decarboxylase (ODC) gene [71].

In an attempt to define a valid non-invasive biomarker to stratify colorectal adenoma risk, Bezawada and colleagues examined the role of PGE-M, the main prostaglandin E2 metabolite in the urine [72]. Results from this study demonstrated that regular use of aspirin reduces the risk of developing advanced, large, and multiple adenomas in patients with high levels of urine PGE-M [72].

Interestingly, within the cohort of Lynch syndrome patients enrolled in the CCFR, Resler and colleagues identified the intronic SNP rs2920421-GA in the ALOX12 gene as a protective variant against CRC development [73].

In 2011, Chan and colleagues investigated the correlation between plasma inflammatory markers and CRC risk and ascertained whether the use of aspirin was differentially associated with the risk of CRC according to the levels of inflammatory markers [74]. The authors found that plasma levels of serum soluble tumor necrosis factor receptor-2, sTNFR-2, but not C-reactive protein (CRP) or interleukin-6 (IL-6), were associated with CRC risk. Intriguingly, aspirin seemed to reduce CRC risk in women with high levels of sTNFR-2 [74]. Another inflammatory marker involved in aspirin response is the macrophage inhibitory cytokine 1 (MIC1). It is involved in the TGFβ pathway and is believed to play a role in CRC carcinogenesis, as higher MIC1 levels were associated with a 93% increased risk of CRC [75]. Interestingly, aspirin users with high plasma levels of MIC1 were found to have a higher risk of developing COX-2-positive CRC [75].

According to Thun and colleagues, daily treatment with low-dose aspirin (75 mg) cannot achieve total inhibition of COX-2 in nucleated cells but causes permanent inhibition of COX-1 in platelets. This, in turn, suppresses the induction of COX-2 in adjacent nucleated cells of the intestinal mucosa at sites of injury during the early stages of tumorigenesis, where platelets are likely to be recruited and activated [76]. The induction of COX-2 leads to reduced apoptosis and increased cell proliferation and angiogenesis. Thus, even low doses of aspirin lead to the downregulation of COX-2 in epithelial and tumor cells [76].

Moreover, it seems that regular aspirin use reduces the risk of COX-2-overexpressing CRCs but not the risk of tumors showing weak or absent COX-2 expression [77]. In 2012, based on the observation that COX-2 inhibition downregulates PI3K activity, Liao and colleagues investigated the role of the PI3K signaling pathway in an attempt to identify potential molecular biomarkers for aspirin chemoprevention. The authors found that regular use of aspirin was associated with longer survival in patients with PIK3CA-mutated CRC [78]. Moreover, the aspirin preventive effect was strongest in patients with tumors showing both PIK3CA mutation and high COX-2 expression [78]. Conversely, BRAF-mutated CRC cells seem to be less sensitive to aspirin’s effects [79]. Indeed, regular aspirin use was associated with a lower risk of tumors with wild-type BRAF and high COX-2 expression. The association between the use of aspirin and decreased risk of BRAF-wild-type CRC was found to be independent of KRAS mutation status [79]. This study suggested that resistance to aspirin in BRAF-mutated cells is due to the upregulation of the MAPK pathway, which results in increased COX-2 and prostaglandin E2 production. These findings were extremely important since Lynch syndrome patients, which are at high risk of CRC, mostly have BRAF-wild-type tumors [80].

Another mechanism of CRC chemoprevention by aspirin involves the inhibition of the WNT pathway. Indeed, aspirin inhibits COX-mediated synthesis of prostaglandin E2, a known activator of β-catenin signaling [81]. β-catenin plays a crucial role in the WNT pathway, and high levels of nuclear β-catenin induce the loss of normal cellular architecture and promote neoplastic conversion [82]. In 2013, Nan and colleagues showed that the SNP rs6983267 on chromosome 8q24 is a CRC susceptibility locus that affects TCF7L2 binding to CTNNB1, the gene encoding β-catenin, thereby affecting its transcriptional activity [83]. A lower CRC risk was observed with the use of aspirin in subjects carrying the T allele of rs6983267, which is associated with reduced expression of the MYC oncogene, the gene most proximate to the SNP. Conversely, the G allele of rs6983267 leads to constitutively active binding of CTNNB1/TCF7L2 and MYC expression, thereby promoting CRC cancerogenesis [84]. According to these data, aspirin chemoprevention could thus be tailored based on the rs6983267 genotype [83].

Considering the important role of hydroxyprostaglandin dehydrogenase 15-(NAD) (15-PGDH) as an antagonist of COX-2 during CRC carcinogenesis, Fink and others hypothesized that susceptibility to aspirin might differ according to 15-PGDH expression levels in the colon mucosa [85]. In the cohorts analyzed by these authors, the regular use of aspirin decreased the risk of CRC only in patients with high 15-PGDH expression [85]. Both aspirin and 15-PGDH reduced CRC risk by decreasing the amount of available prostaglandin. Thus, 15-PGDH levels in normal colon mucosa could be taken advantage of as a marker to identify potential target populations for the use of aspirin as a chemopreventive agent.

In 2015, an interesting study published in Journal of the American Medical Association (JAMA) tested the interaction between the regular use of aspirin and SNPs across the genome in relation to the risk of CRC. The authors identified two different SNPs associated with enhanced benefits from aspirin treatment. In particular, they found an association between regular use of aspirin and reduced risk of developing CRC in individuals carrying the TT genotype of SNP rs2965667, which is located on chromosome 12p12.3, near the microsomal glutathione S-transferase 1 (MGST1) gene, and in individuals carrying the AA genotype of SNP rs16973225, which is located on chromosome 15q25.2, near the interleukin 16 (IL16) gene (Table 2) [86].

Altogether, these findings suggest that for maximum benefit, chemopreventive treatment with aspirin should be tailored, taking into account the status of these (and possibly other) biomarkers.

#### 4.1.2. Non-Aspirin NSAIDs

Non-aspirin NSAIDs (NA-NSAIDs) competitively inhibit both COX-1 and COX-2 [87]. Various NA-NSAIDs have been evaluated for their activity as chemopreventive agents in CRC. Two-month treatment with sulindac (150 mg daily), a non-selective NSAID, was found to reduce the number of ACF lesions both in the general population and in a cohort of individuals with a CRC family history, which represent groups at average and moderate risk of developing CRC, respectively [88]. The benefit of sulindac was also evaluated in a high-risk group consisting of FAP patients. In this population, the use of sulindac (300 mg daily) was associated with a reduction in both the number and size of colonic polyps [89,90,91]. However, after discontinuation of sulindac treatment, an increase was observed in polyp size and number, although at levels that remained statistically lower than baseline [89].

Unfortunately, several studies revealed that the use of non-selective NA-NSAIDs is associated with a high risk of gastrointestinal bleeding, which is probably related to COX-1 inhibition. Thus, it was hypothesized that the use of selective COX-2 inhibitors could be safer [92]. As a result, other NA-NSAIDs able to inhibit COX-2, including celecoxib and rofecoxib, were tested in clinical trials. Celecoxib and rofecoxib were first studied in FAP patients and individually found to cause a regression in polyp number and size compared to placebo [93,94]. Moreover, they were shown to reduce the risk of sporadic colorectal adenoma and CRC in a case-control study performed in the average-risk population [95].

The ability of these specific COX-2 inhibitors to prevent sporadic adenomas was also assessed in three multicenter randomized trials. The first study, entitled Adenoma Prevention with Celecoxib (APC), was launched in 1999. It included 2035 patients treated with twice-daily doses of celecoxib (200 mg or 400 mg) and revealed a dose-dependent, 33–45% reduction in the number of detected sporadic adenomas after 3 and 5 years [96,97]. Moreover, it was found that the genotype influences the dose of celecoxib required to reduce the risk of adenoma. Indeed, high-dose celecoxib was associated with a 5.6% greater reduction in the 3-year cumulative incidence of adenomas compared with low-dose celecoxib in patients carrying the SNP rs1057910-C, a genetic variant of cytochrome P450 2C9 (CYP2C9) (Table 2) [98]. The greater efficacy of high-dose celecoxib in preventing colorectal adenomas thus appears to be confined to individuals with slow metabolizer (CYP2C9*3) genotypes. This suggests that genetic variability influences susceptibility to the potential benefits and risks of celecoxib.

In the second trial, entitled Prevention of Colorectal Sporadic Adenomatous Polyps (Pre-Sap), 400 mg single-dose celecoxib or placebo was given daily to 1561 subjects who had had adenomas removed before enrollment in the study [99]. This randomized, placebo-controlled, double-blind trial demonstrated that the use of 400 mg of celecoxib once daily reduced the occurrence of colorectal adenomas within 3 years after polypectomy [99]. Interestingly, in a subsequent report, the authors observed a statistically significant lower rate of new advanced adenomas in the celecoxib-treated group compared with the placebo group 2 years after the last administration of celecoxib [100].

Finally, the third randomized study, named Adenomatous Polyp Prevention On Vioxx (APPROVe), included 2586 patients with a history of colorectal adenomas, which were treated daily with 25 mg of rofecoxib or placebo [101]. Rofecoxib was found to reduce the formation of adenomas by 24% [102]; however, it was withdrawn from the market by the Food and Drug Administration because of its increased cardiovascular risk [101]. Unfortunately, all NA-NSAIDS, including selective COX-2 inhibitors, are associated with significant side effects, which hinder their use as CRC chemopreventive agents apart from FAP patients due to their high risk of developing CRC [93,103].

#### 4.1.3. 5-Aminosalicylates

As mentioned above, ulcerative colitis is considered a risk factor for developing precancerous lesions and therefore CRC [104]. CRC prevention strategies for ulcerative colitis patients are based on colonoscopy screening; however, the inflammatory background can make the exam more difficult to read [105]. For this reason, the identification of chemopreventive agents amenable to being used in ulcerative colitis patients is crucial. Currently, the aspirin derivatives 5-aminosalicylates (5-ASAs) are the most effective agents for treating ulcerative colitis. In addition, these drugs showed promising results in reducing the risk of CRC in these patients [106]. Indeed, observational studies reported a dose-dependent inverted association between the use of 5-ASAs and the risk of CRC, with the lowest risk being observed when ulcerative colitis patients were treated with at least 1.2 g of mesalamine equivalents per day [107]. In these studies, a lower risk of CRC was detected when 5-ASAs were administered for a minimum of 2–6 months to 20 years. Importantly, pooled results from these observational analyses support the potential use of 5-ASAs as protective agents for CRC in ulcerative colitis patients [106,107,108].

#### 4.1.4. Ursodeoxycholic Acid

Various reports suggest that ursodeoxycholic acid (UDCA), a synthetic bile acid, may also be effective as a CRC chemopreventive agent in ulcerative colitis patients. Starting from in vivo studies, UDCA was shown to be involved in the inhibition of ACF growth and thus prevents the development of CRC [109,110]. Subsequently, a phase III trial in patients with previously diagnosed sporadic adenomas revealed that UDCA treatment reduced the recurrence of adenomas with high-grade dysplasia by almost 40% [111]. Moreover, UDCA was found to reduce both colorectal dysplasia and CRC in patients with ulcerative colitis and associated primary sclerosing cholangitis [112,113].

### 4.2. Metabolic Agents

Since type 2 diabetes and hypercholesterolemia are both considered risk factors for CRC [114,115], various metabolic agents have been evaluated as chemopreventive factors.

#### 4.2.1. Metformin

Metformin is an insulin-sensitizing drug belonging to the biguanide class and is commonly prescribed to patients with type 2 diabetes. Several reports revealed that metformin also has an anti-tumor effect, including on CRC, in both diabetic and non-diabetic patients [116,117,118,119]. Studies performed in mice showed that the use of metformin reduced ACF and adenomas in azoxymethane-treated animals and decreased polyp generation in the APC^Min/+^ CRC mouse model [120,121].

Interestingly, in 2010, a short-term randomized study in non-diabetic patients showed that metformin reduced by 40% the formation of CRC precancerous lesions such as ACF compared to controls [122]. These findings recently prompted the investigators to perform another RCT. Patients with both colorectal ACF and resectable polyps were recruited and treated with aspirin (100 mg) and/or metformin (250 mg) for 8 weeks, after which polypectomy was performed to evaluate changes in the number of ACF. The final results of this trial are still pending [123]. In another report, the use of low-dose metformin (250 mg daily) for one year was associated with a 40% lower incidence of adenomas and a 33% reduced number of total colon polyps without side effects in non-diabetic patients after polypectomy [124]. Moreover, a meta-analysis of ten different studies revealed an inverse association between metformin treatment and colorectal adenoma risk and colorectal tumors in patients with diabetes and also showed a trend in lowering the risk of adenomas in non-diabetic patients with a history of adenomas or CRC [125].

RCTs support the use of metformin mainly in diabetic patients at high risk of CRC and, to a lesser extent, in other populations at moderate and high risk; however, epidemiology data are not always consistent. Indeed, while most epidemiology studies showed decreased CRC risk in metformin users [126,127,128,129,130,131,132,133], some reports failed to detect an association between metformin treatment and the risk of developing adenomas or CRC [134,135,136,137]. Nevertheless, based on a systematic review and meta-analysis of metformin intake and CRC mortality, metformin was found to improve survival in CRC patients with diabetes [138].

Overall, the evidence gathered to date suggests that the use of metformin should be considered in clinical practice as a chemoprevention strategy for CRC, especially in diabetic patients.

#### 4.2.2. Statins

Statins are inhibitors of 3-hydroxy-3-methylglutaryl coenzyme A (HMG-CoA) reductase and are commonly prescribed for their lipid-lowering properties [139]. Of note, HMG-CoA was found overexpressed in several CRC cell lines, and in vitro studies showed that the use of statins inhibits cell proliferation and increases apoptosis [140,141]. These observations were confirmed in vivo in mice with chemically or genetically induced colorectal neoplasia, in which the use of statins alone or with NSAIDs reduced the development of CRC [142,143,144]. However, inconsistent results were gathered from studies investigating the effect of statins on colorectal adenoma incidence in patient cohorts [145,146]. A population-based case-control study, entitled the Molecular Epidemiology of Colorectal Cancer Study, was conducted in northern Israel from 1998 to 2004 in patients previously diagnosed with CRC. This observational study revealed that the use of statins for more than 5 years was associated with a 47% decrease in the risk of developing CRC [147]. Moreover, two different clinical trials showed that the use of pravastatin or simvastatin reduced the number of new cases of CRC by 43% and 19%, respectively, over a 5-year follow-up period [148,149]. However, multiple meta-analyses failed to detect an association between statin use and CRC risk [150,151,152,153]. Interestingly, genetic variation may influence the effect of statins on CRC risk. Indeed, Lipkin and colleagues observed a significant association between statin intake and a reduced risk of developing CRC in individuals carrying the AA genotype of SNP rs12654264 in the HMGCR gene (Table 2) [154].

Considering their ease of administration, safety, tolerability, and low cost, along with promising preclinical data, statins could offer substantial benefits as chemopreventive agents; however, clinical studies provided conflicting results. Thus, further trials are needed to ascertain the potential of statins in CRC chemoprevention.

#### 4.2.3. Long-Chain Omega-3 Polyunsaturated Fatty Acids

Long-chain omega-3 polyunsaturated fatty acids (PUFAs) are important nutrients involved in decreasing inflammation and are primarily found in dark-meat fish [155]. In agreement with the role of inflammation in CRC carcinogenesis, fish consumption was associated with a reduced risk of developing CRC [156,157]. In particular, two different PUFAs, EPA and docosahexaenoic acid (DHA), were shown to have anti-neoplastic effects in in vivo studies [158,159,160]. These results were confirmed in a cohort of patients at high risk of CRC. Indeed, an RCT in FAP patients revealed that EPA intake (500 mg twice daily for 6 months) significantly reduced the number and size of rectal adenomas [161]. Consistent data were obtained in a secondary analysis of the previously mentioned multicenter, randomized seAFOod trial, which found that EPA treatment was associated with a decrease in left-sided and conventional adenomas [43].

Moreover, in a prospective cohort study in the average-risk general population, individuals taking fish oil supplements (4 or more days/week) for 3 or more years had about 50% lower risk of developing CRC, with greater benefits for colon cancer than for rectum cancer and in men compared to women [162]. However, other meta-analyses of pooled results from epidemiological and prospective cohort studies failed to report a significant association when comparing the highest and lowest doses of fish or PUFAs [163,164,165].

Despite the attractive safety and tolerability profiles of long-chain omega-3 PUFAs such as EPA and DHA, additional studies are needed to demonstrate their benefits in reducing colorectal adenoma and CRC risk.

#### 4.2.4. Folic Acid

Folic acid (or folate) is a micronutrient abundantly found in fruits and vegetables. It is believed to potentially contribute to CRC chemoprevention by maintaining the normal DNA methylation pattern needed for DNA synthesis and repair [166,167]. Various studies showed that several factors, including the dosage of folic acid intake, may affect its role as a preventive or promoting agent in CRC cancerogenesis [168,169]. Indeed, modest levels of folate supplementation appear to suppress cancer development, while high doses seem to enhance it [66]. Further evidence suggests that folate protects against adenoma formation but promotes the progression of existing colorectal neoplasia [170]. In vivo studies provided conflicting results showing a correlation between folate deficiency and a reduced risk of developing CRC [166,171]. On the other hand, epidemiology studies found an association between the intake of folic acid and a decreased risk of both colorectal adenomas and CRC [172,173,174], while a meta-analysis of 8 RCTs failed to reveal a significant association between folate treatment and adenoma recurrence in both high-risk and average-risk patient populations [175]. Conversely, two different investigations involving women cohorts (the NHS and the Canadian National Breast Screening Study) indicated that supplementation with folate was protective against CRC, with dose-dependent benefits [176,177].

Further reports indicate that folate intake may be important in preventing the development of CRC in patients with ulcerative colitis. Indeed, case-control studies revealed that daily folate supplementation may reduce the risk of ulcerative colitis-related CRC [178,179].

The involvement of folate in CRC carcinogenesis has also been suggested by additional findings. For example, it has been reported that folate deficiency may induce TP53 mutation, with a low intake of folate being associated with an increased risk of TP53-mutated CRC [180]. Moreover, polymorphisms that affect the activity of methylenetetrahydrofolate reductase (MTHFR) may modify individual cancer risk [181]. MTHFR plays an important role in folate metabolism by contributing to the maintenance of circulating levels of folate and methionine, thereby preventing the accumulation of homocysteine [182]. Various meta-analyses confirmed that homozygosity for the MTHFR C677T polymorphism (Table 2) is associated with a significantly reduced risk of developing CRC [181,183,184].

Overall, there is no strong evidence that folate is an effective chemopreventive agent for CRC; thus, further investigations are needed to support its potential use in CRC chemoprevention.

### 4.3. Antioxidants

Oxidative stress plays a major role in mutagenesis, carcinogenesis, and aging. As a result, in recent years, there has been great interest in the potential health benefits of dietary and antioxidant supplements for cancer prevention.

#### 4.3.1. Selenium

Selenium is an essential cofactor for the antioxidant enzyme glutathione peroxidase, which protects against oxidative damage to lipids, lipoproteins, and DNA [185,186]. The effect of selenium supplementation on CRC risk was reviewed by three different meta-analyses of RCTs.

Bjelakovic and colleagues included in their analysis several studies investigating various antioxidant agents, including selenium. While none of the other antioxidants revealed beneficial effects, selenium seemed to potentially reduce gastrointestinal cancers and associated mortality, although these observations could be influenced by the low methodological quality of most of the assessed trials. Indeed, only one of the trials in which selenium was given as a single antioxidant had a low-bias risk [187].

In 2011, Papaioannou and colleagues published a systematic review and meta-analysis to assess the available evidence on the clinical effectiveness of antioxidants (vitamins A, C, E, selenium, and β-carotene) for the prevention of adenomas and/or CRC in the general population [188]. Their findings failed to show a positive effect associated with an increased intake of antioxidants, including selenium [188].

By contrast, in another meta-analysis published in 2013, selenium seemed to show promising results. Indeed, the authors found that it was the only antioxidant having a positive effect on CRC risk reduction. In particular, selenium supplementation (200 μg/day) was associated with a trend in lowering both colorectal adenoma recurrence (RR = 0.70) and CRC incidence (RR = 0.88). Importantly, selenium intake was also related to decreased overall mortality (RR = 0.91) [189]. 

As for its side effects, selenium supplementation has been associated with a statistically significant (*p* < 0.01) increase in alopecia and grade 1–2 dermatitis [190].

#### 4.3.2. Vitamins A, C, E and β-Carotene

Since antioxidants such as vitamins A, C, E, and β-carotene are involved in reducing oxidative stress by neutralizing free radicals, their intake from the diet has been investigated in various reports. A pooled analysis of prospective cohort studies found that the total intake of vitamins C and E from the diet was associated with a modest decrease in CRC risk, while no correlation was found for vitamin A or β-carotene [191]. However, a meta-analysis of 12 RCTs evaluating the effects of vitamins A, C, and E as well as other compounds, concluded that these agents were not effective as chemopreventive agents for CRC in the general population [188].

The potential benefits of vitamin A were also assessed in two different meta-analyses of observational studies [192,193]. In one of these analyses, a significant association between vitamin A intake and a reduced risk of developing colon cancer was found when comparing individuals taking vitamin A to non-users [192].

Three different studies investigating the effect of vitamin C on CRC prevention were published from 2011 to 2015. All of them failed to show a protective role for this vitamin [188,192,193].

The potential benefits of vitamin E as a CRC chemopreventive agent were evaluated in seven different meta-analyses published between 2007 and 2015. Also in this case, none of the studies could detect a significant effect of vitamin E intake on CRC risk [187,188,189,192,193,194,195].

In addition, several meta-analyses evaluating the effects of β-carotene supplementation (alone or in combination with other agents) on CRC risk were published from 2004 to 2013. Again, no significant association between β-carotene consumption and primary prevention of CRC was detected in any of these studies [187,188,189,196,197].

Overall, the available evidence indicates that intake of these antioxidants is not associated with a reduction in the risk of developing CRC.

#### 4.3.3. Curcumin

Curcumin is a phytochemical derived from turmeric (*Curcuma longa*). It is commonly used as a dietary supplement and is renowned for its antioxidant properties; therefore, its intake has been advocated for chemopreventive, anti-metastatic, and anti-angiogenic purposes [198].

Two studies were performed to assess the role of curcumin in adenoma chemoprevention in FAP patients [199,200]. In the first study, FAP patients treated orally with 480 mg of curcumin and 20 mg of quercetin 3 times a day for at least 6 months showed a 60% reduction in polyp number and a 50% decrease in polyp size at endoscopy [199]. However, in the second study, a double-blind randomized trial, no difference was found in the mean number or size of lower intestinal tract adenomas in FAP patients treated with curcumin at 3000 mg/day for 1 year compared to placebo [200].

Based on these conflicting results, the potential chemopreventive effect of curcumin, alone or in association with other agents, will have to be assessed in larger patient cohorts.

### 4.4. Minerals and Vitamin D

Recently, dietary supplements have been studied as chemopreventive agents. This strategy is known as the ‘nutraceutical’ approach. Nutraceuticals are products that contain vitamins and minerals as main ingredients and are consumed in different forms, such as tablets, capsules, powder, or soft gels. This approach has the advantage of virtually no harmful side effects [201].

#### 4.4.1. Magnesium

Magnesium is an essential mineral for several processes involved in CRC carcinogenesis, including DNA synthesis and repair, cell proliferation, and apoptosis. In vivo studies carried out about 30 years ago revealed that magnesium intake has a preventive effect on CRC in rat models [202,203]. More recently, a meta-analysis of 8 prospective studies found that higher magnesium intake seemed to be associated with a modest reduction in the risk of CRC and, in particular, colon cancer, with a pooled relative risk of 0.81 and 0.94 for colon and rectal cancer, respectively [204]. A concomitant report including a case-control study on colorectal adenomas (768 cases; 709 polyp-free control subjects) and a meta-analysis of colorectal adenomas (3 case-control studies) and carcinomas (6 prospective cohort studies) provided similar results. Indeed, in the case-control study, inverse associations between magnesium intake and the risk of colorectal adenomas were only observed in subjects with a BMI ≥ 25 kg/m^2^ or older than 55 years and for advanced adenomas, while in the meta-analysis, increased magnesium intake was associated with a lower risk of colorectal adenomas and CRC [205]. Of note, in a meta-analysis of epidemiologic studies, evaluating the correlation between dietary magnesium intake and the risk of all cancers, higher magnesium intake showed a significant preventive effect only in CRC, especially in female participants [206].

Altogether, these findings suggest that magnesium intake is a promising approach in CRC chemoprevention; however, important factors such as optimal dosage, appropriate indications, and potential toxicity need to be further investigated in future studies with larger samples.

#### 4.4.2. Calcium

Calcium is believed to protect against CRC by binding bile acids and fatty acids within the lumen of the colon or by directly inhibiting cell proliferation [207,208,209]. Its potential role as a chemopreventive agent was first confirmed in in vivo studies showing that calcium supplementation reduced CRC development in mouse models [210]. Subsequently, the Calcium Polyp Prevention Study Group trial showed that patients with prior adenomas taking 3 g of calcium daily were less prone to develop other adenomas [211]. This effect was later confirmed for advanced adenomas [212] and was shown to continue for at least 5 years after discontinuation of calcium intake [213]. However, another study investigating individuals with prior adenomas, entitled the European Cancer Prevention Intervention Study, demonstrated that the intake of 2 g/day of calcium did not induce a significant decrease in adenoma recurrence in this moderate-risk population [214]. Further evidence emerged from a systematic review and meta-analysis performed by Carroll and colleagues in 2010 [215]. The authors analyzed RCTs evaluating the efficacy of calcium intake in reducing colorectal adenomas and CRC risk in populations at average, moderate, and high risk. Their findings showed that calcium intake, with or without vitamin D, did not affect the relative risk of CRC in populations at average risk, whereas a statistically significant reduction in the relative risk of adenoma recurrence was observed in individuals at moderate risk (with a history of adenomas). On the other hand, FAP patients, which are a population at high risk, did not benefit from supplemental calcium, as shown by the number of developed adenomas [215].

Interestingly, evidence from observational studies suggested that CRC risk decreased upon calcium supplementation [216,217], although some positive results were limited to the distal colon and the rectum [218,219]. However, in the WHS trial, in which participants received 500 mg of calcium carbonate and 200 IU of vitamin D twice daily for seven years, no notable differences in CRC incidence were observed compared to placebo-treated patients [220].

Although a modest protective effect of calcium intake emerged from observational studies, conflicting results were reported in meta-analyses of RCTs. Thus, the efficacy of calcium supplementation as a chemopreventive strategy needs further investigation.

#### 4.4.3. Vitamin D

Vitamin D plays an important role in calcium metabolism and is also involved in other physiological functions. It has been shown to reduce cell proliferation, inhibit angiogenesis, and promote cell differentiation, which are mechanisms through which it may reduce the risk of CRC [221,222]. Moreover, it may be effective as a chemopreventive agent because activated vitamin D receptors have been found to repress β-catenin signaling [8]. Additional data from in vivo studies also indicated that vitamin D has anti-inflammatory activity [223,224]. Various RCTs and case-control studies designed to evaluate the role of 25(OH)-vitamin D revealed an inverse association with colorectal adenoma, CRC, and rectal cancer [220,225,226]. Moreover, the NHS study showed that the relative risk of CRC decreased linearly across quintiles of plasma 25(OH)-vitamin D concentration, with an almost 50% risk reduction for the highest compared to the lowest quintile [227]. Similarly, the Women’s Health Initiative (WHI) study showed an inverse correlation between 25(OH)-vitamin D levels and CRC relative risk [220]. However, the final results of this study gathered after 7 years of follow-up did not support the use of vitamin D as a chemopreventive agent to reduce the risk of developing CRC. According to the investigators, this could be due to various reasons. First of all, 7 years might have not been sufficient to reveal an effect on cancer incidence since CRC tumorigenesis is a process that occurs over decades. Furthermore, the vitamin D dose of 400 IU/day administered to study participants might have been too low to yield significant results [220]. However, similar results were obtained in a subsequent trial carried out with higher doses of vitamin D (2000 IU/day) associated or not with omega-3 (1000 mg/day) vs. placebo, in which the investigators found no difference in CRC incidence among the different patient groups [228].

In summary, meta-analyses, observational studies, and clinical trials evaluating the use of vitamin D, alone or in association with calcium, to reduce CRC incidence provided conflicting results. Larger clinical trials are thus needed to further ascertain the potential benefits associated with the use of vitamin D as a CRC chemopreventive agent.

### 4.5. Hormone Replacement Therapy

The observation that pre-menopausal women are much more protected against CRC than postmenopausal women prompted researchers to investigate the role of hormones in reducing CRC risk [229]. Estrogens may act against CRC through different mechanisms, including reduced production of insulin-like growth factor-I (IGF-1) or secondary bile acids [230,231]. Based on this evidence, the correlation between hormone replacement therapy and CRC risk has been evaluated in several prospective studies, most of which showed an inverse association between hormone use and the risk of both colorectal adenomas [232,233,234,235] and CRC [236,237,238,239]. These outcomes were also confirmed in various clinical trials. For example, in the Women Health Initiative (WHI) study, the combination of estrogen and progestin was found to reduce the risk of developing CRC by almost 40% [240]. However, estrogen alone did not prove effective in reducing CRC risk [241,242], a result that was also validated in other studies [243,244]. It was hypothesized that progestin enhances the estrogenic effect of conjugated estrogen, making combined therapy more biologically active than estrogen alone in the colon [244]. Interestingly, a case-control study in which participants were stratified for microsatellite instability (MSI) showed that combined hormone therapy with estrogen and progestin was associated with a statistically significant (about 40%) reduction in CRC risk in women with MSI-low or MSI-stable tumors, while no clear correlation was found in women with MSI-high tumors (Table 2) [243].

Despite the benefits that may be associated with hormone replacement therapy in the reduction in CRC risk, potential side effects should also be considered. Indeed, it has been shown that postmenopausal hormones increase the risk of breast cancer and cardiovascular events, and thus the risk-benefit profile must be carefully evaluated [242].

### 4.6. Dietary Products

International comparisons of tumor incidence suggest that people consuming a Western diet are at an increased risk of CRC [75]. Based on this observation, nutritional chemoprevention based on fiber, whole grains, fruits, and vegetables has been proposed as a potential strategy to reduce the risk of developing CRC. One of the possible mechanisms involved in the protective effect of these products is that increased dietary fiber intake accelerates the transit of lumen contents, thereby decreasing the exposure of colonic cells to carcinogens [245]. Five different meta-analyses of observational studies published from 1990 to 2018 found that increased whole grain consumption was associated with a lower risk of developing colon cancer and CRC, with a significant dose effect [246,247,248,249,250]. Further meta-analyses of observational studies published from 1990 to 2017 investigated the role of fruits and vegetables in CRC risk. Most but not all of these studies revealed that fruits and vegetables have a protective effect against CRC, with a relative risk ranging from 0.48 to 0.92 [163,246,248,251,252,253,254,255,256]. However, only vegetable intake (100 g/day) showed a significant inverse association with CRC risk in a linear dose-response analysis. Interestingly, patients who were previously consuming low amounts of fruits and vegetables showed the highest CRC risk reduction after increasing their intake [251].

Dietary products such as fiber, fruits, and vegetables meet all the prerequisites of the ideal chemopreventive agent; thus, it would be very important to gather further evidence on their effects and optimal intake strategy in populations at average, moderate, and high risk of developing CRC.

### 4.7. Vaccine Strategy

In the last few years, advances in medical sciences have led to the use of vaccines for CRC immunoprevention, especially in high-risk patients. Patients with a hereditary genetic syndrome predisposing to CRC are considered the ideal population for the development of a preventive vaccine because the involved mutations are predictable, resulting in a specific group of neoantigens that can be directly targeted by vaccines [257]. There are two types of antigens that can be incorporated into cancer vaccines: tumor-specific antigens (TSAs) and tumor-associated antigens (TAAs) [258]. TSAs are directly generated by somatic mutations in tumor cells; as such, they are not expressed in normal cells.

Lynch syndrome patients carry indel mutations in microsatellites of coding genes, which can result in the synthesis of frameshift peptides (FSP) [258]. FSPs are tumor-specific neoantigens shared across patients with MSI. Recently, a clinical trial was conducted to investigate vaccines based on a combination of three recurrent FSPs (TAF1B (−1), HT001/ASTE1 (−1), AIM2 (−1)), all derived from indel mutations in coding microsatellite regions [259]. Interestingly, all participants receiving the vaccine developed humoral and T-cell responses against the neoantigens, and no severe vaccine-related side effect was identified. This trial demonstrated for the first time that vaccines can also be used for preventing MMR-deficient cancers [259].

Subsequently, another TSA vaccine was developed based on FSPs shared across patients with MSI. This viral-vectored vaccine, encoded by 209 selected FSPs, was named Nous-209. Nous-209 immunogenicity was first demonstrated in mice with potent and broad induction of FSP-specific CD8 and CD4 T-cell responses. Then, the authors demonstrated in vitro that the vaccine was processed by human antigen-presenting cells and was subsequently able to activate human CD8 T-cells. These findings demonstrate the feasibility of a vaccine for CRC prevention [260]. Currently, the Nous-209 vaccine is being studied in two ongoing clinical trials (NCT04041310 and NCT05078866). The first one was launched in 2019 and is designed to detect evidence of anti-tumor activity of the Nous-209 vaccine plus pembrolizumab combination therapy in adults with unresectable or metastatic dMMR or MSI-H CRC [261]. The second one is being conducted to evaluate the safety and effect of the Nous-209 vaccine in Lynch syndrome patients [262]. No result has been posted yet for either study.

In an effort to achieve a CRC preventive vaccine strategy also for FAP patients, ERBB3, a pseudo-kinase member of the EGFR/ERBB family of receptor tyrosine kinases, was targeted by using a synthetic peptide vaccine in APC^Min/+^ mice [263]. In this study, the development of humoral and cellular immunity was associated with a reduced number of polyps in vaccinated animals [263].

In populations with a moderate risk of developing CRC, such as individuals diagnosed with premalignant lesions in the colon, a strategy based on TAA vaccines provided promising results. In a phase I/II open-label study, Kimura and colleagues tested a peptide vaccine based on the TAA MUC1 in subjects with a history of advanced colorectal adenoma. The vaccine consisted of 100 µg of a MUC1 100 mer peptide mixed with 500 µg of Hiltonol^®^, an adjuvant toll-like receptor 3 agonist. The vaccine was administered at weeks 0, 2, and 10. To assess memory response, a booster dose was given at week 52 [264]. This MUC1-derived vaccine was well tolerated and capable of inducing long-term immunological memory [264]. More recently, this vaccine was used in a randomized, double-blind, placebo-controlled, multicenter trial in individuals with prior adenomas. The primary outcome was adenoma recurrence at the first colonoscopy >1 year after the initial vaccination. The authors observed a nearly 40% reduction in adenoma recurrence in participants who had an immune response at week 12 and with the booster injection [265].

Based on these encouraging results, further clinical studies are warranted to evaluate the efficacy of peptide vaccines to prevent CRC malignant progression.

### 4.8. Target Therapy

Increasing interest has been directed toward the use of target therapy compounds, alone or in combination with other treatments such as NSAIDs, as chemopreventive agents for CRC. The most promising compound in this class is difluoromethylornithine (DFMO), which is an irreversible inhibitor of ornithine decarboxylase, the enzyme that catalyzes the conversion of ornithine to putrescine. This reaction is the first step in polyamine synthesis, which is implicated in cell proliferation [266]. Interestingly, it has been found that colorectal adenomas and CRC have increased levels of polyamines compared to normal colon mucosa [267]. Based on this observation, DFMO was evaluated, alone or in combination with other compounds, as a potential CRC chemopreventive agent. A randomized, placebo-controlled, double-blind trial showed that DFMO (500 mg) and sulindac (150 mg) administered once daily markedly reduced recurrent colorectal adenomas with few side effects [268]. The efficacy of this combination was later compared with either drug alone in a high-risk population consisting of FAP patients. It was found that the incidence of disease progression in FAP patients was not significantly lower with the combined treatment compared with DFMO or sulindac alone [269]. In another randomized trial, FAP patients were treated with DFMO in combination with celecoxib [270]. The combined treatment reduced the number of adenomas by 13%; however, no significant difference in adenoma count was observed compared to celecoxib alone. Conversely, synergistic effects in the reduction in CRC risk were observed for DFMO with cyclosporine and selenium [271,272]. However, it should be noted that a common side effect associated with DFMO is ototoxicity [270].

A large body of evidence supports the role played by the epidermal growth factor receptor (EGFR) in CRC tumorigenesis [273]. Indeed, EGFR has been found overexpressed in up to 50% of CRCs [274]. As a result, efforts to target EGFR in order to reduce the risk of CRC are ongoing [275]. Since in vitro and in vivo studies showed that the EGFR signaling pathway is involved in COX-2 expression [276], a trial evaluating combined treatment with the non-specific COX-2 inhibitor sulindac 150 mg twice daily and the EGFR tyrosine kinase inhibitor erlotinib 75 mg daily vs. placebo was performed in FAP patients. The combination of sulindac and erlotinib was found to lower the colorectal polyp burden after 6 months of treatment [277]. Interestingly, a secondary analysis of this trial revealed that the reduction in polyp burden occurred both in patients with an entire colorectum and in patients with only a rectal pouch or rectum [278].

Since interleukin 23 (IL-23) has been shown to sustain CRC progression [279], a further trial on the high-risk group of FAP patients was conducted using guselkumab, an anti-IL-23 monoclonal antibody. However, this study is ongoing, and no results have been published yet [280].

**Table 1 ijms-24-07597-t001:** Chemopreventive agents for CRC. Abbreviations: 5-ASAs, 5-aminosalicylates; ACF, aberrant crypt foci; AIM2, Absent In Melanoma 2; AMPK, 5-adenosine monophosphate-activated protein kinase; AOM, azoxymethane; APACC, Association pour la Prévention par l’Aspirine du Cancer Colorectal; APC, Adenoma Prevention with Celecoxib; APPROVe, Anomatous Polyp Prevention on Vioxx; ASCOLT, Aspirin for Dukes C and High-Risk Dukes B Colorectal Cancers; ASPIRED, Aspirin Intervention for the Reduction of CRC Risk; ATP, adenosine 5′-triphosphate; CALGB, Cancer and Leukemia Group B; CAPP, Colorectal Adenoma/Carcinoma Prevention Program; CCFR, Colon Cancer Family Registry; COX, cyclooxygenase; CPS II, Cancer Prevention Study II; CRC, colorectal cancer; DFMO, difluoromethylornithine; EB3IV, recombinant protein ERBB3 residues 269–396; EBX, extracellular amino acid residues 299–323 of ERBB3; EGFR, epidermal growth factor receptor; EPA, eicosapentanoic acid; ERBB3, Erb-B2 Receptor Tyrosine Kinase 3; FAP, familial adenomatous polyposis; FSP, frameshift peptides GAd, Great Ape Adenovirus; HMG-CoA, hydroxy-β-methylglutaryl-CoA; HPFS, Health Professionals Follow-up Study; KLH, keyhole limpet hemocyanin; IL-23, interleukin-23; HT001, protein asteroid homolog 1; MUC-1, mucin-1; MVA, Modified Vaccinia virus Ankara; N.A., not applicable; NA-NSAIDs, non-aspirin non-steroidal anti-inflammatory drugs; NHS, Nurses’ Health Study; PHS, Physicians’ Health Study; ppm, parts per million; Pre-Sap, Prevention of Colorectal Sporadic Adenomatous Polyps; PUFA, polyunsaturated fatty acid; RB, retinoblastoma; RCT, randomized controlled trial; seAFOod, Systematic Evaluation of Aspirin and Fish Oil; TAA, tumor-associated antigen; TF1B, TATA-Box Binding Protein Associated Factor, RNA Polymerase I Subunit B; UDCA, ursodeoxycholic acid; ukCAP, United Kingdom Colorectal Adenoma Prevention; USPSTF, US Preventive Services Task Force; WHS, Women’s Health Study.

Agent	Primary Target	Mechanism	Endpoint	Study or Trial (Years)	Participants (n)	Age of Participants	CRC Risk Level	Dose	Median Time of Follow-Up	Results	Ref
**Anti-inflammatory agents**											
Aspirin	COX-1 and COX-2 (irreversible inhibition)	Inhibits prostaglandin synthesis and the β-catenin WNT pathway	ACF	In vivo studies	AOM-treated rats	N.A.	N.A.	0.2–0.6%	N.A.	Reduced ACF number and size	[35,36]
			Adenoma	CALGB (1993–2000)	Individuals with prior CRC (517)	30–80 years	High	325 mg daily	13 months	Reduced adenoma risk	[38]
				CAPP1 (1993–2005)	FAP patients (133)	10–21 years	High	600 mg twice daily	After 1 year and then annually	Reduced adenoma largest size	[41]
				AFPPS (1994–1998)	Individuals with prior adenomas (1121)	21–80 years	Moderate	81 mg or 325 mg daily	3 years	Low dose but not high dose reduced the risk of adenoma recurrence	[37]
				ukCAP (1997–2005)	Individuals with prior adenomas (945)	Younger than 75 years	Moderate	300 mg daily	3 years	Reduced adenoma recurrence risk	[39]
				APACC (1997–2001)	Individuals with prior adenomas (272)	18–75 years	Moderate	160 mg or 300 mg daily	1 and 4 years	Reduced adenoma risk after 1 year, but not after 4 years	[40,42]
				seAFOod (2010–2017)	Individuals with prior adenomas (709)	55–73 years	Moderate	300 mg daily	1 year	Reduced number of conventional and serrated adenomas in the right colon at secondary analysis	[43]
				Cross-sectional studies (2011–2014)	General population divided into smokers and non-smokers and people with CRC family history (2918)	45–65 years	Average and moderate	81 mg	30 months	Reduced adenoma risk only in non-smoker users	[45]
			CRC	NHS (1980–2000)	General population (82,911 female)	30–55 years	Average	325 mg 2 times per week	Every 2 years	Reduced CRC risk	[46]
				PHS (1982–1995)	General population (22,071 male)	40–84 years	Average	325 mg on alternate days	5–12 years	No reduced CRC incidence	[47]
				CPS II (1982–1988)	General population (662,424)	57 years (mean)	Average	100 mg on alternate days	10–18 years	Reduced CRC risk	[48]
				HPFS (1986–1990)	General population (47,900 male)	40–75 years	Average	100 mg 2 times per week	2–4 years	Reduced CRC risk and metastatic CRC	[49,50]
				WHS (1993–2004)	General population (39,876 female)	45 years or older	Average	100 mg on alternate days	1-10-18 years	Reduced CRC incidence only after 10 or 18 years of follow-up	[51,52]
				CCFR (1997–2012)	Lynch syndrome patients (1858)	43 years (mean)	High	Twice a week	Not reported	Reduced CRC risk	[53]
				CAPP2 (2001–2008)	Lynch syndrome patients (861)	45 years (mean)	High	600 mg daily	2–10 years	No reduced CRC risk after two years of follow-up, a strong reduction in CRC risk at 10 years	[54,55]
				USPSTF (2004–2015)	General population and individuals with prior adenomas	40–79 years	Average and moderate	75 mg daily or on alternate days	10–20 years	Reduced CRC risk and mortality	[56,57]
				J-CAPP (2007–2012)	Individuals with prior adenomas (311)	40–70 years	Moderate	100 mg daily	2 years	Reduced CRC risk in the non-smoker population	[58]
				ASCOLT (2008-ongoing)	Dukes’ B and C CRC (1587)	18 years and older	High	200 mg daily	Every 3 months for 3 years + every 6 months for another 2 years	Final results pending	[59]
				Pooled analysis derived from 4 RCTs and 1 study of different doses of aspirin (2010)	General population (13,500)	45–69 years	Average	75 mg or 300 mg daily	20 years	Reduced CRC risk and mortality	[60]
				ASPIRED (2010-ongoing)	Individuals with prior adenomas (180)	50–69 years	Moderate	81 mg or 325 mg daily	Every 6 months	Final results pending	[61]
				CAPP3 (2014–2019)	Lynch syndrome patients (1500)	Not reported	High	100 mg or 300 mg or 600 mg daily	5 years	Final results pending	[62]
				J-FAPP (2015–2017)	FAP patients (311)	40–70 years	High	100 mg and/or mesalazine daily	8 months	Reduced adenoma and CRC risk	[58]
NA-NSAIDs	COX-1 and COX-2 (reversible inhibition)	Inhibit prostaglandin synthesis and WNT signaling pathway	ACF	Sulindac	General population and individuals with a CRC family history (304)	55–75 years	Average and moderate	150 mg	2 months and 1 year	Reduced ACF number	[88]
			Adenoma	Sulindac	FAP patients (46)	14–46 years	High	300 mg daily	1 year	Reduced adenoma risk	[89,90,91]
				Double-blind, placebo-controlled study (celecoxib) (1996–1998)	FAP patients (77)	18–65 years	High	100 mg or 400 mg twice daily	6 months	Reduced adenoma risk	[93]
				Double-blind, placebo-controlled study (rofecoxib)	FAP patients (21)	Not reported	High	25 mg	3-6-9 months	Reduced adenoma risk	[94]
				Nested case-control study (rofecoxib and celecoxib)	General population (3477)	65 years or older	Average	Not reported	3 months	Reduced adenoma risk	[95]
				APC trial (1999–2002)	Individuals with prior adenomas (2035)	31–88 years	Moderate	Celecoxib 200 mg or 400 mg twice daily	3–5 years	Reduced adenoma risk	[96,97]
				Pre-Sap (2001–2005)	Individuals with prior adenomas (1561)	30 years or older	Moderate	Celecoxib 400 mg daily	1–3 years	Reduced adenoma risk	[99]
				APPROVe (2001–2004)	Individuals with prior adenomas (2586)	40–96 years	Moderate	Rofecoxib 25 mg daily	1–3 years	Reduced adenoma risk	[101]
5-ASAs	Derivatives of aspirin	Inhibit prostaglandin synthesis	Adenoma and CRC	Observational studies (1972–2002)	Ulcerative colitis patients	Not reported	High	Mesalamine >1.2 g/day Sulfasalazine >2.4 g/day	10-20-30 years	Reduced adenoma and CRC risk	[106,107,108]
UDCA	Secondary bile acids	Disruption of the balance between colorectal crypt cell proliferation, differentiation, and apoptosis	ACF	In vivo studies	AOM-treated Fisher male rats (344)	N.A.	N.A.	UDCA 0.2% or 0.4% for 2 weeks	28 weeks	Reduced ACF number	[109,110]
			Adenoma and CRC	Phase III clinical trial	Individuals with prior adenomas and ulcerative colitis patients (1285)	40–80 years	Moderate and high	300 mg	3 years	Reduced adenoma and CRC risk	[111]
				Cross-sectional study	Ulcerative colitis and primary sclerosing cholangitis patients (59)	Not reported	High	9.9 mg/kg daily	3 years	Reduced adenoma and CRC risk	[112,113]
**Metabolic agents**											
Metformin	Inhibits mitochondrial complex I to prevent the production of mitochondrial ATP	Activates AMPK, reduces cyclin D1 expression and RB phosphorylation	ACF and adenoma	In vivo studies	AOM-BALB/c mice	N.A.	N.A.	250 mg/kg daily	6 weeks for ACF and 32 weeks for adenomas	Reduced ACF and adenoma risk	[120]
			Adenoma	In vivo studies	APC^Min/+^ mice	N.A.	N.A.	250 mg/kg	N.A.	Reduced number of intestinal polyps larger than 2 mm	[121]
			ACF, adenoma	Short-term randomized study	Non-diabetic patients (26)	65–75 years	Average	250 mg daily	1 month	Reduced ACF and adenoma risk	[122]
			ACF	RCT	Non-diabetic patients (60)	Not reported	Average	250 mg and/or aspirin 100 mg daily	8 weeks	Final results pending	[123]
			Adenoma	Multicenter double-blind, placebo-controlled, randomized phase 3 trial (2011–2014)	Non-diabetic patients (498)	20 years or older	Average	250 mg daily	1 year	Reduced prevalence and number of metachronous adenomas or polyps after polypectomy	[124]
			Adenoma and CRC	Case-control studies and RCT (2008–2016)	Non-diabetic and diabetic patients, individuals with prior adenomas and CRC (8726)	40–89 years	Average, moderate, and high	≥250 mg daily	4–15 years	Reduced adenoma and CRC risk	[125]
				Epidemiology studies	Non-diabetic and diabetic patients	20–80 years	Average and high	250 mg or 500 mg daily	1–3 years	Conflicting results	[126,127,128,129,130,131,132,133,134,135,136,137]
			CRC	Retrospective cohort study	Diabetic patients (60,520)	40 years or older	High	750–4000 mg daily	5 years	Reduced CRC risk	[138]
Statins	HMG-CoA reductase (reversible inhibition)	Disruption of the mevalonate pathway	Adenoma and CRC	In vivo studies	APC^Min/+^ mice	6-week-old	N.A.	Pitavastatin at doses of 20 and 40 ppm	14 weeks	Reduced adenomas in a dose-dependent way	[142]
				In vivo studies	AOM-treated F344 rats	5-week-old	N.A.	Atorvastatin 100–200 ppm and/or sulindac 100 ppm or naproxen 150 ppm	45 weeks	Reduced CRC risk	[143]
				In vivo studies	APC^Min/+^ mice	6-week-old	N.A.	Atorvastatin 100 ppm and/or celecoxib 300 ppm	80 days	Reduced adenoma and CRC risk	[144]
			Adenoma	Review of endoscopy and pathology databases	Individuals with prior adenomas (2626)	63 years (mean)	Moderate	Not reported	3–5 years	Reduced adenoma risk	[145]
				Secondary analysis of data from three large colorectal adenoma chemoprevention trials	General population (2915)	Not reported	Average	Not reported	Not reported	No reduced adenoma risk	[146]
			CRC	Molecular Epidemiology of Colorectal Cancer Study (1998–2004)	Individuals with prior CRC (3968)	58–80 years	High	Not reported	5 years	Reduced CRC risk	[147]
				Double-blind trial	Patients with myocardial infarction who had plasma total cholesterol levels below 240 mg/dL and low-density lipoprotein (LDL) cholesterol levels of 115 to 174 mg/dL (4159)	50–70 years	Moderate	Pravastatin 40 mg daily	5 years	Reduced CRC risk	[148]
				Survival study	Patients with angina pectoris or previous myocardial infarction and serum cholesterol levels of 5.5 to 8.0 mmol/L (4444)	35–70 years	Moderate	Simvastatin 20–40 mg daily	5 years	Reduced CRC risk	[149]
				Systematic review and meta-analysis	General population	40–80 years	Average	Not reported	3–6 years	Conflicting results	[150]
Long-Chain Omega-3 PUFAs	Components of phospholipids that form cell membranes	Anti-proliferative, apoptotic, and anti-angiogenic properties	ACF	In vivo studies	Wistar rats	N.A.	N.A.	EPA 18.7%; DHA 8%	48 h	Reduced ACF number	[158]
			ACF, adenoma, and CRC	In vivo studies	APC^Min/+^ mice, AOM-treated mice, xenograft mice	N.A.	N.A.	EPA 4–16%; DHA 0.75–6%	1 day-32 weeks	Reduced ACF number, adenoma, and CRC risk	[159]
			Adenoma	Prospective study (2006–2007)	FAP patients (55)	18 years or older	High	EPA 500 mg twice daily	6 months	Reduced adenoma risk	[161]
				seAFOod (2010–2017)	Individuals with prior adenomas (709)	55–73 years	High	EPA 2 g daily	1 year	Reduced number of conventional and left-sided adenomas at secondary analysis	[43]
			CRC	Prospective study (2000–2008)	General population (68,109)	50–76 years	Average	Fish oil more than 4 days per week	3 years	Reduced CRC risk	[162]
				RCTs (2001–2011)	General population, FAP patients	40–75 years	Average and high	EPA 0.09 vs. 0.03 g daily DHA 0.18 vs. 0.08 g daily	3–22 years	Conflicting results	[163,164,165]
Folic acid	Coenzyme in single transfers in the synthesis of nucleic acid and amino acid metabolism	Maintaining normal DNA methylation required for synthesis and repair	ACF and CRC	In vivo studies	AOM-treated rats (159)	6-week old	N.A.	0, 2, 5, or 8 mg/kg	34 weeks	Conflicting results	[166,171]
			Adenoma and CRC	Epidemiology studies	General population	Not reported	Average	100 μg or 600 μg daily	Not reported	Reduced CRC risk	[172,173,174]
			Adenoma	RCT	General population, individuals with prior adenoma	65 years (mean)	Average and high	0.5 to 2.5 mg daily	36–88 months	No reduced adenoma risk	[175]
			CRC	NHS (1980–1994)	General population (88,756 female)	30–55 years	Average	200 μg or 400 μg daily	Every 2 years	Reduced risk of CRC	[176]
				Canadian National Breast Screening Study	General population (5681)	Not reported	Average	200 μg or 400 μg daily	10 years	Reduced risk of CRC	[177]
				Case-control studies	Ulcerative colitis patients	Not reported	High	0.4–1.0 mg daily	Not reported	Reduced risk of CRC	[178,179]
**Antioxidant agents**											
Selenium	Trace minerals required to make selenium-containing proteins	Antioxidant properties	Adenoma and CRC	RCT	General population	62 years (mean)	Average	200 μg daily	6–12 years	Conflicting results	[187,188,189]
Vitamin A	Combines with retinol-binding protein	Regulates nuclear receptors that are involved in tumor formation	CRC	Observational studies	General population	34–80 years	Average	1 μg daily	8–10 years	Conflicting results	[192,193]
Vitamin C	Cofactor in collagen formation and tissue repair	Reduces oxidative stress	CRC	RCT	General population	40–80 years	Average	75 mg or 250 mg or 500 mg daily	5–9 years	No reduced CRC risk	[188,192,193]
Vitamin E	Primarily ends up in cell and organelle membranes	Inhibits lipid peroxidation in membranes	CRC	RCT	General population	Not reported	Average	30 mg or 50 mg or 600 mg daily	6–12 years	No reduced CRC risk	[187,188,189,192,193,194,195]
β-carotene	Functions as a provitamin A	Antioxidant properties	CRC	RCT	General population	55 years (mean)	Average	20 mg or 30 mg daily	2–12 years	No reduced CRC risk	[187,188,189,196,197]
Curcumin	Inhibits reactive oxygen-generating enzymes	Antioxidant properties	Adenoma	Prospective study	FAP patients (5)	Not reported	High	Curcumin 480 mg and quercetin 20 mg orally 3 times a day	Every 3 months	Reduced adenoma risk	[199]
				RCT (2011-2016)	FAP patients (44)	18–85 years	High	3000 mg daily	1 year	No reduced adenoma risk	[200]
**Minerals and vitamin D**											
Magnesium	Involved in metabolism, insulin resistance, and inflammation	Important for DNA synthesis and repair	ACF and CRC	In vivo studies	Methylazoxymethanol acetate-treated male F344 rats	N.A.	N.A.	250 ppm or 500 ppm 1000 ppm	4-6-8 weeks	Reduced ACF and CRC risk	[202,203]
			CRC	In vivo studies	Methylazoxymethanol acetate-treated male F344 rats	N.A.	N.A.	500 ppm or 1000 ppm	227 days	Reduced CRC risk	[203]
			CRC	Prospective studies (2005–2012)	General population (338,979)	40–75 years	Average	50 mg daily	8–28 years	Reduced CRC risk	[204]
			Adenoma	Case-control studies	General population, individuals with a CRC family history	18–75 years	Average and moderate	100 mg daily	Not reported	Reduced adenoma risk	[205]
			Adenoma and CRC	Epidemiologic and prospective studies	General population (1,236,004)	Not reported	Average	300–400 mg daily	Not reported	Reduced adenoma and CRC risk	[206]
Calcium	Incorporated into the skeleton	Bile acid-binding capacity	CRC	In vivo studies	1,2-Dimethylhydrazine (DMH)-treated Slac mice (80)	N.A.	N.A.	1.24–3.0%	24 weeks	Reduced CRC risk	[210]
			Adenoma	Calcium Polyp Prevention Study Group RCT	Individuals with prior adenomas (930)	61 years (mean)	Moderate	3 g daily	1-4-9 years	Reduced advanced adenoma recurrence risk	[211,212,213]
				The European Cancer Prevention Intervention Study	Individuals with prior adenomas (665)	35–75 years	Moderate	2 g daily	3 years	No significant effect on adenoma risk	[214]
				Systematic review and meta-analysis of RCTs (2010)	General population, individuals with prior adenomas, FAP patients	16–80 years	Average, moderate, and high	500 mg^−2^ g^−3^ g daily	6 months–7 years	No positive results for average- and high-risk populations, reduced adenoma risk in individuals with a history of adenomas	[215]
			CRC	Cancer Prevention Study II Nutrition Cohort (1992-1993)	General population (1,277,499)	50–74 years	Average	500 mg daily	5 years	Reduced CRC risk	[216]
				Prospective study (2000)	General population (61,463)	53 years (mean)	Average	900 mg daily	12 years	Reduced CRC risk	[217]
				NHS and HPFS	General population (135,342)	30–75 years	Average	500–1250 mg daily	10–16 years	Reduced distal colon cancer risk	[218]
				Prospective study	General population (34,702)	Not reported	Average	Not reported	9 years	Reduced rectal cancer risk	[219]
				WHS	General population (36,282 female)	50–79 years	Average	Calcium carbonate 500 mg and vitamin D 200 IU twice daily	7 years	No reduced CRC risk	[220]
Vitamin D	Regulates gene transcription by binding vitamin D receptors	Inhibits proliferation and angiogenesis	Adenoma, CRC, and rectal cancer	RCT	General population, individuals with prior adenomas	50–79 years	Average and moderate	400 IU daily	7 years	Conflicting results	[220,225,226]
			CRC	RCT	General population (25,871)	50 years or older	Average	Vitamin D 2000 IU and omega-3 fatty acids 1 g daily	5 years	No reduced CRC risk	[227,228]
**Hormone replacement therapy**											
Hormones	Increase the production of insulin-like growth factor-I or secondary bile acids	Inhibit proliferation and promote cell cycle arrest and apoptosis	Adenoma	Prospective studies	Individuals with prior adenomas (411)	30–74 years	Moderate	Not reported	Not reported	Reduced adenoma risk	[232,233,234,235]
			CRC	The Molecular Epidemiology of Colorectal Cancer Study (1998–2006)	Individuals with prior CRC (1234)	60 years or older	High	Not reported	5 years	Reduced CRC risk	[236,237,238,239]
				Women’s Health Initiative (WHI) RCT	General population (postmenopausal status) (10,739)	50–79 years	Average	Conjugated equine estrogen 0.625 mg plus medroxyprogesterone acetate 2.5 mg daily	7 years	No reduced CRC risk	[240,241,242]
**Dietary products**											
Fibers	Involved in the metabolism and catabolism of bioactive food components	Decrease the exposure of colonic cells to carcinogens	CRC	RCT	General population	25–76 years	Average	90 g daily increments	6–16 years	Reduced CRC risk	[246,247,248,249,250]
Fruits and vegetables	Involved in the metabolism and catabolism of bioactive food components	Decrease the exposure of colonic cells to carcinogens	CRC	RCT	General population	34–82 years	Average	100 g daily increments	Not reported	Reduced CRC risk	[163,246,248,251,252,253,254,255,256]
**Vaccines**											
FSP-based vaccines	TAF1B(−1), HT001(−1), and AIM2(−1)	Development of humoral and T-cell responses against FSPs	CRC	Phase I/IIa clinical trial (2011–2015)	Lynch syndrome (22)	55 years (mean)	High	3 cycles of subcutaneous vaccinations mixed with Montanide ISA-51 VG	6 months	Enhanced immune response against FSP peptides	[259]
Nous 209 viral-vectored vaccine	209 FSPs	Neoantigen-based vaccine for the treatment of MSI tumors	Immunogenic response	In vivo studies	CB6F1 mice	6-week-old	N.A.	GAd-209-FSP and MVA-209-FSP were administered i.m. at the dosage of 4 × 10^8^ vp and 4 × 10^7^ ifu, respectively	3 weeks	CD8 and CD4 T-cell responses	[260]
			CRC	Phase I/II clinical trial (2019–2025)	Individuals with prior CRC (34)	18 years or older	High	GAd-209-FSP low dose; MVA-209-FSP low dose; GAd-209-FSP high dose; MVA-209-FSP high dose; GAd20-209-FSP; RP2D; MVA-209-FSP, RP2D	Up to 110 weeks	Final results pending	[261]
				Phase Ib/II clinical trial (2021–2025)	Lynch syndrome patients (45)	18 years or older	High	GAd-209-FSP and MVA-209-FSP	Every 12 months	Final results pending	[262]
Synthetic peptide	ERBB3	Development of humoral and cellular immunity against FSPs	Adenoma	In vivo studies	APC^Min/+^ mice	3-week-old	N.A.	100 mg of EBX peptide, EB3IV, or KLH in 100 mL of a 50/50 mixture of antigen and CFA	3 months	Reduced recurrent adenomas	[263]
TAA vaccine	MUC-1-derived peptides	Anti-MUC-1 IgG response	Adenoma	Phase II clinical trial—RCT (2008–2013)	Individuals with prior adenomas	40–70 years	Moderate	100 µg MUC1 + Hiltonol^®^at week 0, 2, 10, and 52	54 weeks	Reduced recurrent adenomas	[264,265]
**Target therapy**											
DFMO	Ornithine decarboxylase (irreversible inhibition)	Inhibits polyamine synthesis	Adenoma	RCT	Individuals with prior adenomas (375)	40–80 years	Average and high	DFMO 500 mg daily and/or sulindac 150 mg	36 months	Reduced recurrent adenomas	[268]
				RCT	FAP patients (171)	18 years or older	High	DFMO 750 mg and/or sulindac150 mg	48 months	Conflicting results	[269]
				RCT	FAP patients (112)	38 years (mean)	High	DFMO 250 mg and/or celecoxib 400 mg	6 months	Modest reduction in adenoma risk	[270]
Erlotinib	EGFR tyrosine kinase inhibitor (reversible inhibition)	Inhibits EGFR signaling	Adenoma	RCT (2010–2014)	FAP patients (92)	41 years (mean)	High	Erlotinib 75 mg daily and/or sulindac 150 mg twice daily	6 months	Reduced recurrent adenomas	[277,278]
Guselkumab	Monoclonal antibody against IL-23 subunit alpha	Inhibits IL-23 signaling	Adenoma	RCT	FAP patients	Not reported	High	Not reported	Not reported	Final results pending	[280]

**Table 2 ijms-24-07597-t002:** Molecular biomarkers associated with CRC chemopreventive agent effects. Abbreviations: 15-PGDH, 15-hydroxyprostaglandin dehydrogenase; APC, Adenoma Prevention with Celecoxib; CCFR, Colon Cancer Family Registry; COX-2, cyclooxygenase 2; CRC, colorectal cancer; GSEC, Genetic Susceptibility to Environmental Carcinogens; GWAS, genome-wide association study; HPFS, Health Professionals Follow-up Study; MECC, Molecular Epidemiology of Colorectal Cancer; MIC1, macrophage inhibitory cytokine 1; MSI, microsatellite instability; NHS, Nurses’ Health Study; PGE-M, prostaglandin E2 major metabolite; PI3KCA, phosphatidylinositol-4,5-bisphosphate 3-kinase catalytic subunit alpha; RCT, randomized controlled trial; SNP, single-nucleotide polymorphism; sTNFR-2, serum soluble tumor necrosis factor receptor-2; UGT1A6, UDP glucuronosyltransferase family 1 member A6.

Biomarker	Type	Chemopreventive Agent	Study (Number of Participants)	Endpoint	Description	Outcome	Ref
rs2070959-G	Genetic variant	Aspirin	NHS (1062)	CRC	G genotype SNP	Protective against CRC	[69,70]
rs4365457-C	Genetic variant	Aspirin	NHS (1062)	CRC	C genotype SNP	Protective against CRC	[69,70]
rs2430420-GG	Genetic variant	Low-dose aspirin	AFFPS nested cohorts within an RCT (370)	Adenoma	GG genotype SNP	Protective against CRC	[71]
rs28362380-TT	Genetic variant	Low-dose aspirin	AFFPS nested cohorts within an RCT (370)	Adenoma	TT genotype SNP	Protective against CRC	[71]
PGE-M	Urine levels	Aspirin	NHS and AFPPS (748)	Adenoma	High levels	Protective against CRC	[72]
rs2920421-GA	Genetic variation	Aspirin	CCFR (1621)	CRC	GA genotype SNP	Protective against CRC	[73]
sTNFR-2	Plasma levels	Aspirin	NHS (280)	CRC	High levels	Protective against CRC	[74]
MIC1	Plasma levels	Aspirin	NHS and HPFS (618)	CRC	High levels	Promotes COX-2-positive CRC	[75]
COX-2	Overexpression in tumor	Aspirin	NHS and HPFS (632)	CRC	High levels	Protective against CRC	[76,77]
PIK3CA	Mutation in tumor	Aspirin	NHS and HPFS (2190)	CRC	PIK3CA exons 9 and 20 mutated in tumor	Protective against CRC	[78]
BRAF	Mutation in tumor	Aspirin	NHS and HPFS (1226)	CRC	BRAF V600E mutated in tumor	Promotes CRC	[79]
rs6983267-T	Genetic variant	Aspirin	NHS and HPFS (840)	CRC	T genotype SNP	Protective against CRC	[83,84]
15-PGDH	Colon mucosa levels	Aspirin	NHS and HPFS (270)	CRC	High levels	Protective against CRC	[85]
rs2965667-TT	Genetic variant	Aspirin	GWAS (8,634)	CRC	TT genotype SNP	Protective against CRC	[86]
rs16973225-AA	Genetic variant	Aspirin	GWAS (8,634)	CRC	AA genotype SNP	Protective against CRC	[86]
rs1057910-C	Genetic variant	Celecoxib	APC (2,035)	Adenoma	C genotype SNP	Protective against CRC	[98]
rs12654264-AA	Genetic variant	Statins	MECC (4,187)	CRC	AA genotype SNP	Protective against CRC	[154]
MTHFR 677TT	Genetic variant	Folic acid	GSEC (30,650)	CRC	TT genotype SNP	Protective against CRC	[181,182,183]
MSI	Microsatellite instability	Hormone replacement therapy (estrogen and progestin)	Case-control studies	CRC	MSI-low or stable tumors	Protective against CRC	[243]

## 5. Conclusions

The identification of which subgroups of individuals will most likely benefit from and should thus be treated with chemopreventive agents is one of the priorities of preventive medicine; however, no conclusive data have been gathered to date to answer this crucial question in the field of CRC. Indeed, many aspects have to be considered, including CRC risk level, potential side effects, genetic factors such as SNPs or other variants, and the determination of intermediate endpoints such as ACF or colorectal adenomas.

While some of the chemopreventive agents discussed in this review are currently recommended for subjects with increased risk of genetic syndromes predisposing to CRC, such as Lynch syndrome or cardiovascular diseases, chemoprevention largely remains an unmet need for populations at average and moderate risk of developing CRC. Some natural and synthetic compounds have actually shown promising results in these populations and may reach clinical practice in the near future, yet further studies are needed to find compounds that can meet all the requirements of the ideal chemopreventive agent, i.e., efficacy, safety, tolerability, low cost, wide availability, and ease of administration.

## Figures and Tables

**Figure 1 ijms-24-07597-f001:**
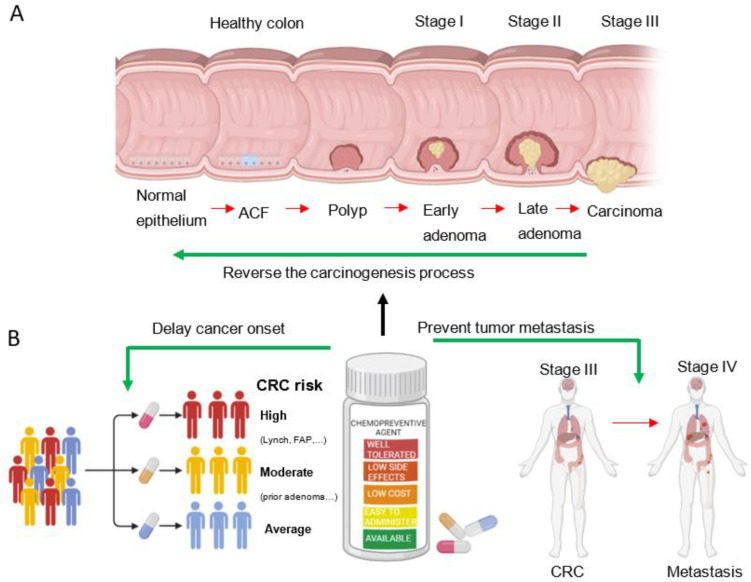
CRC tumorigenesis and chemoprevention. (**A**) Schematic representation of the adenoma-carcinoma sequence, which includes the progression steps, such as aberrant crypt foci (ACF) and adenoma lesions, that are used as endpoints in clinical trials to evaluate potential chemopreventive agents. (**B**) Chemopreventive agents are natural and synthetic compounds intended to delay cancer onset, reverse the carcinogenesis process, and prevent tumor recurrence and metastasis. The ideal chemopreventive agent should be well tolerated, safe, easy to administer, and readily available at low cost. Moreover, chemopreventive agents should be tailored to individuals at high (carriers of genetic syndromes predisposing to CRC, such as Lynch syndrome, all different types of FAP syndrome, MUTYH-associated polyposis, Peutz-Jeghers syndrome, juvenile polyposis syndrome, Cowden syndrome, and hamartoma tumor syndrome, patients with diabetes mellitus), moderate (subjects with a prior diagnosis of colonic adenoma or a family history of CRC), or average (general population, with particular concern for non-Hispanic Black men and individuals with an unhealthy diet) risk of developing CRC (**B**). Created with Biorender.com.

## Data Availability

Not applicable.

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
