# Peer review of "Colorectal Cancer Chemoprevention: A Dream Coming True?"

_ijms, 2023, doi:10.3390/ijms24087597_

Round 1

Reviewer 1 Report

In this manuscript the authors overviewed and systematized existing to date chemopreventive agents for colorectal cancer, both natural and synthetic. The presented review covers a wide range of drugs and ongoing research in the world on this topic. It is not surprising that the Conclusions section raises more questions than answers.

Author Response

We thank the Reviewer for this comment.

Reviewer 2 Report

Good review of literature.

Section of ASA far too long, especially compared to all other discussed chemopreventive agents

Inadequate discussion of vaccine strategy for chemoprevention, especially in high risk patients

Author Response

We are grateful to the Reviewer for these advices. We agree to the Reviewer that the ASA section is not well balanced with the rest of manuscript. However, aspirin is currently recommended for subjects with increased risk of genetic syndromes predisposing to CRC such as Lynch syndrome, or cardiovascular diseases, so by now aspirin is the main agent used for CRC chemoprevention and that has actually moved from bench to bedside. Moreover, as suggested by the Reviewer, we introduce the paragraph about the vaccine strategy in CRC chemoprevention.

Reviewer 3 Report

The authors (Signorile, et.al.) of this article entitled "Colorectal cancer chemoprevention: a dream coming true?" describe the evaluation of several compounds as chemopreventive agents for colorectal cancer in a number of clinical trials. The article is well written and overall easy to read. There is only a minor correction needs to be made: Figure 1 hides the text and the latter is not readable. 

Author Response

We are grateful to the Reviewer for pointing out this inaccuracy. In this amended version of the manuscript we have fixed the minor concern by moving the figure.  
